# Functional Characterization of Serum Amyloid P Component (SAP) in Host Defense against Bacterial Infection in a Primary Vertebrate

**DOI:** 10.3390/ijms23169468

**Published:** 2022-08-22

**Authors:** Jiadong Li, Hao Bai, Xiaoxue Yin, Zhelin Wu, Li Qiu, Xiayi Wei, Qingliang Zeng, Liangliang Mu, Jianmin Ye

**Affiliations:** 1Guangdong Provincial Key Laboratory for Healthy and Safe Aquaculture, Guangzhou Key Laboratory of Subtropical Biodiversity and Biomonitoring, School of Life Sciences, South China Normal University, Guangzhou 510631, China; 2Guangdong Laboratory for Lingnan Modern Agriculture, Guangzhou 510642, China

**Keywords:** *Oreochromis niloticus*, SAP, bacterial infection, phagocytosis, hemolysis

## Abstract

Serum amyloid P component (SAP), an ancient short pentraxin of the pentraxin family, plays an essential role in resistance to bacterial infection. In this study, the expression and functional characterization of SAP (OnSAP) in Nile tilapia (*Oreochromis niloticus*), a primary vertebrate, are investigated. The open reading frame of *OnSAP* is 645 bp of a nucleotide sequence encoding a polypeptide of 214 amino acids. As a calcium-binding protein, the structure and relative motif of OnSAP is highly similar to those of humans, containing amino acid residues Asn, Glu, Gln and Asp. In healthy fish, *OnSAP* mRNA is extensively distributed in all eleven tissues examined, with the highest level in spleen. The mRNA expression of *OnSAP* was significantly up-regulated after being challenged with gram-positive bacterium *Streptococcus agalactiae* and gram-negative bacterium *Aeromonas hydrophila* in vivo. In addition, recombinant OnSAP ((r)OnSAP) protein had capacities of binding *S. agalactiae* or *A. hydrophila* in the presence of Ca^2+^. Further, (r)OnSAP helped monocytes/macrophages to efficiently phagocytize bacteria. Moreover, the (r)OnSAP was able to enhance the complement-mediated lysis of the chicken red blood cells. Collectively, the evidence of SAP in tilapia, based on the results including its evolutionary conserved protein structure, bacterial binding and agglutination, opsonophagocytosis of macrophage and hemolysis enhancement, enriches a better understanding of the biological functions of the pentraxin family.

## 1. Introduction

Innate immunity plays an important role in resistance against pathogens, consisting of cellular and humoral arms [1]. Soluble pattern recognition molecules (PRMs) are parts of the humoral arm that recognize pathogen-associated molecular patterns (PAMPs), and regulate and sustain the immune response in coordination with the cellular arm [2,3]. Collectins, ficolins and pentraxins, as the PRMs in immune system, are able to recognize pathogens, activate the complement system, regulate inflammation, and so on [4].

Pentraxins, including short-chain pentraxins such as C-reactive protein (CRP/PTX1), serum amyloid P component (SAP/PTX2) and long-chain pentraxins, are PRMs of the immune system, constituting a conservative protein superfamily characterized by a cyclic multimeric structure [5,6]. Protein motifs of pentraxins are characterized by the presence of a 200-amino acid PTX domain in the C-terminal and an intramolecular conserved amino acid sequence named pentraxin signature (*HxCxS/TWxS*, x representing any amino acid). As for long-chain pentraxins, for example, pentraxin member 3 (PTX3), has an additional N-terminal domain preceding the PTX domain [7], whereas others have only a PTX domain in short-chain pentraxin, including CRP and SAP [8].

SAP is a key member of pentraxins with a PTX domain and an intramolecular conserved pentraxin signature, which could form an oligomer by the interaction of five identical units, playing a critical role in innate immune defense [9]. As a Ca^2+^-dependent protein, SAP is mainly secreted from the hepatocytes and macrophages into the circulation [10]. SAP is able to bind not only viruses (e.g., influenza), but also bacteria (e.g., *Streptococcus pneumoniae*, *Staphylococcus aureus*, *Escherichia coli*, *Streptococcus pyogenes* and *Neisseria meningitidis*) and parasites such as malaria [1,11]. Besides, SAP makes leukocyte-mediated phagocytosis by interacting with the D1 and D2 domains of FcγRIIa contact the ridge helices from two neighboring SAP subunits, which could be inhibited by competitive inhibition by IgG due to a shared binding site, resulting in the inhibition of immune complex-mediated phagocytosis [12,13]. Moreover, SAP could activate the complement system by interacting with C1q, C4-binding protein (C4BP) and MBL, contributing to the enhancing host defense [6,11,14,15].

In teleost fish, the SAP has been reported in a few species, including rainbow trout (*Oncorhynchus mykiss*), Atlantic salmon (*Salmo salar*), half-smooth tongue sole (*Cynoglossus semilaevis*) and rock bream (*Oplegnathus fasciatus*) [2,16,17,18]. In trout, SAP native protein was isolated and identified from the serum [16]. In addition, in response to *Escherichia coli* LPS and *Aeromonas salmonicida*, the dynamics of trout SAP level were investigated [17]. Besides, expression of SAP in half-smooth tongue sole and rock bream was found increased after bacterial and viral pathogen infection [2,18]. Existing in species such as horseshoe crabs, reptiles, lizards and mammals, SAP acts as an ancient immune mediator. However, the distribution and function of SAP might be different due to different species living in different environments [7]. Although some studies focusing on SAP have been reported, research on SAP related to bacterial infection such as phagocytosis, regulation of complement system and so on in teleost fish remains unclear. Thus, a more comprehensive study is urgently needed.

In this study, the functional characterization of SAP was examined and identified in Nile tilapia (*Oreochrmis niloticus*), a primary vertebrate. The ORF of *O*. *niloticus* SAP (*OnSAP*) gene was identified and analyzed. The mRNA expression of OnSAP in different tissues of healthy or stressed fish has been explored. Further, functions of recombinant OnSAP protein towards bacteria binding, agglutination and promotion of phagocytosis were characterized. Moreover, we also found that the OnSAP could facilitate complement-mediated cell lysis. These findings give important evidence that OnSAP is likely to involve in host defense against pathogen infection in a primary vertebrate.

## 2. Results

### 2.1. Primary Sequence Analysis of OnSAP

The obtained ORF sequence of *OnSAP* was 645 bp and encodes for 214 amino acid residues, including a 21-amino acid-length signal peptide and a remaining part named PTX domain (Figure 1A). A structure named Pentraxin signature (*SICSTWDS*) was located in the middle of the protein, which is similar to other species. As for the similarities among sequences, OnSAP shares a 56.73%, 47.98%, 47.27%, 36%, 38.12%, 37%, and 37.37% sequence similarity with *Oryzias melastigma* SAP, *Oncorhynchus kisutch* SAP, *Cynoglossus semilaevis* SAP, *Chelonia mydas* SAP, *Mauremys reevesii* SAP, human SAP and mouse SAP, respectively, revealing its evolutionary conservation (Figure 1B,D). Additionally, the intramolecular disulfide bond was also conservative, but *Cys90* was located in Pentraxin signature (Figure 1B). Further, information on the phylogenetic tree indicated that the SAP of *O. niloticus* firstly was clustered with *Oryzias melastigma*, *Kryptolebias marmoratus* et al., and then clustered with another branch, including *O. kisutch*, *O. mykiss* et al., which formed a branch representing bony fish. Similarly, SAP of *H. sapiens* clustered with *Chlorocebus sabaeus*, a primate, and this branch with another branch including *M. musculus*, *Cricetulus migratorius*, forming a new branch representing mammals. The above results reflect the evolutionary route of the SAP protein (Figure 1C).

### 2.2. The Expression Profiles of OnSAP

*OnSAP* is widely expressed in 11 examined tissues according to tissue distribution analysis. *OnSAP* gene is highly expressed in spleen, liver and heart. The most abundant level of *OnSAP* mRNA in the spleen was 150 times higher compared with that in muscle, which was the lowest expression tissue (Figure 2A).

Stimulated by bacteria, expression of *OnSAP* transcript was significantly up-regulated. In head kidney, after 12 h challenged by *Aeromonas. hydrophila,* LPS and *S. agalactiae,* there was a 200-fold, a 40-fold and 4-fold increase in expression, respectively (Figure 2B). In spleen, *A. hydrophila* and LPS shared similar trends that after all-time points except 12 h, the expression on *OnSAP* increased almost 100-fold. However, as for *S. agalactiae,* the expression was generally downgraded except for 48th h (Figure 2C). In peripheral blood, after being stimulated by *A. hydrophila* and LPS for 12 h, the *OnSAP* was around 2- and 40-fold higher than that of the PBS group, respectively. However, the expression was generally downgraded after being stimulated by *S. agalactiae* (Figure 2D).

### 2.3. Recombinant OnSAP Expression and Purification

Recombinant OnSAP protein ((r)OnSAP) was purified following the protocol of Ni-NTA Agarose Resin (Novagen, Germany). According to SDS-PAGE analysis, (r)OnSAP had a mass weight of about 40 kDa (Figure 3A). Identified by Western blotting analysis, the recombinant protein reacts positively (Figure 3B) to the anti-6 × His-tag mouse monoclonal antibody or polyclonal mouse anti-(r)OnSAP antibody (Figure 3C). Excluding the 21 kDa molecular weight of Trx vector protein (Trx) [19], the mass weight of (r)OnSAP is about 20 kDa, demonstrating the successful preparation of OnSAP, as predicted.

### 2.4. Tridimensional Structure Analysis of OnSAP

Three-dimensional model of OnSAP from different angles was shown in Figure 3D, where pLDDT (LDDT), which displayed the confidence of per-residue in the range 0–100, is best used for intra-domain confidence. Besides, Predicted Aligned Error (PAE) is an index for determining between domain or between chain confidence [20]. According to these indexes, OnSAP structure prediction is accurate; in detail, the pLDDT remained above 90 (confidence is defined as ‘very high’ when over 90) at the start of the prediction (Figure 3E), whereas PAE stayed below 10 (Figure 3F). Appendix A showed Ramachandran PLOT analysis provided by UCLA-DOE LAB (saves.mbi.ucla.edu/ (accessed on 20 July 2022)), which contains the information that the proportion of the most favored regions residues is 93%. Figure 3G displayed an alignment between OnSAP and one of the monomers of human SAP. The Root Means Square (RMS) Displacement between these proteins is 0.942. The calcium-binding site of OnSAP is Ile-52, Asn-53, Glu-132, Gln-133 and Asp-134, whereas, of human SAP, the calcium-binding site is Asp-58, Asn-59, Glu-136, Gln-137 and Asp-138, which shows the high similarity. Moreover, we also compared the similarity among other pentraxins of tilapia. Figure 3H displayed the structure alignment of OnSAP-OnCRP (left) and OnSAP-OnPTX3 (right). RMSD of OnSAP (blue)-OnCRP (pink) is 0.33 and RMSD of OnSAP (blue)-OnPTX3 (green) is 1.528. Interestingly, OnSAP was similar to the PTX domain of OnPTX3. In conclusion, these proteins are structurally conserved so that their function that follows their structure might be similar.

### 2.5. Binding and Agglutinating Activity of (r)OnSAP to Bacteria

The binding between (r)OnSAP protein and bacteria was evaluated by the ELISA assay. In the presence of Ca^2+^, (r)OnSAP had the ability to bind *S. agalactiae* (Figure 4A) and *A. hydrophila* (Figure 4B) in a protein concentration-dependent situation. However, the binding was not detected in the (r)Trx protein or TBS group.

Agglutinating activity of (r)OnSAP was detected under the microscope in the absence of Ca^2+^ using FITC-labeled *S. agalactiae* and *A. hydrophila.* As shown in Figure 4C (a,b,e,f), different concentrations of (r)OnSAP could agglutinate *S. agalactiae* and *A. hydrophila*. However, results of the (r)Trx and TBS group indicated that there was no agglutination capability in the control groups for both bacteria.

### 2.6. Enhancement of Phagocytosis by (r)OnSAP

Effect of (r)OnSAP regulation towards phagocytosis of MO/Mø was detected by flow cytometer. An increase in phagocytosis rate occurred while evaluating the MO/Mø phagocytosing (r)OnSAP-bacteria complex. For details, the MO/Mø phagocytosing percentage was almost 0 and almost no fluorescent signal is released without the existence of bacteria (Figure 4D,E Grey Phenogram). MO/Mø treated with TBS and (r)Trx demonstrated almost the same fluorescent signal (Figure 4D,E Black and Blue Phenogram) and phagocytosing percentage (Figure 4F,G) towards *A. hydrophila* and *S. agalactiae*. After being treated by (r)OnSAP, the fluorescent signal (Figure 4D,E Pink Phenogram) and phagocytosing percentage (Figure 4F,G) was significantly increased in both bacteria investigated. These results indicate that (r)OnSAP is able to promote the phagocytosis of bacteria cells by monocytes/macrophages.

### 2.7. Enhancement of Tilapia Complement-Mediated Cell Lysis by (r)OnSAP

To prove that OnSAP could stimulate complement-mediated hemolysis, (r)OnSAP, (r)Trx, normal tilapia serum (using 50 μL per tube based on CH50 analysis) and CRBCs were carried out. The results demonstrated that (r)OnSAP significantly improved the CRBCs lysis compared with the TBS and (r)Trx groups. Moreover, by adding highly concentrated (r)OnSAP at the same time, the hemolysis rate was almost close to the total hemolysis rate (Figure 4H).

## 3. Discussion

Serum amyloid P component is a soluble pattern recognition molecule and is also known as acute-phase protein, which makes it possible to recognize various pathogenic microorganisms, activate the complement system, join the process of opsonophagocytosis and regulate the inflammatory reaction [21]. In mammals, SAP is a major acute phase of mice while in humans, it is a stable plasma protein [22]. Moreover, SAP is also essential to help humans against *A. fumigatus*, *S. pneumoniae* and *Candida albicans* by activating the complement system and promoting phagocytosis [6,11,23]. However, in teleost fish, there are very few comprehensive studies involving SAP [18].

This study was the first time to realize the functional characterization of OnSAP, a consecutive protein containing a PTX domain and a conservative but special pentraxin signature (*SICSTWDS*), which contains conserved cysteine residues that form intramolecular disulfide bonds (Figure 1B) [24]. Sharing similar protein sequences as well as tridimensional structures with other species, (r)OnSAP is a valuable protein to explore the similarities and differences of PTX family in different species.

As the tissue distribution analysis shows, *OnSAP* mRNA was highly expressed in spleen and liver, and was also widely expressed in all tissues we studied in this study (Figure 2A). This result was a little different from the finding in humans where SAP was mainly produced by hepatocytes [1]. However, for fish such as *P. platessa* [2], high levels of expression occur in brain and gills, and in the rock bream, head kidney and trunk kidney were the high expression tissues [18], which indicated that different species living in a different environment has their special expression patterns.

To explore whether OnSAP gets involved in host resistance to pathogens, expression patterns in vivo were evaluated. Firstly, *OnSAP* transcription of three tissues including spleen, head kidney and peripheral blood stimulated by LPS and bacteria such as *S. agalactiae* and *A. hydrophila* was detected (Figure 2B–D). By the way, *S. agalactiae* and *A. hydrophila* are the main bacterial pathogens of tilapia [25]. The result indicated that after the *A. hydrophila* and LPS challenge, the transcript of *OnSAP* was significantly increased and displayed similar trends, which enriched the yin-yang function of the SAP towards LPS and gram-negative bacterium [26]. Although the response to *S. agalactiae* was significantly upregulated only at specific time points in specific tissues, the results indicated that the synthesis and secretion of OnSAP might be highly relative with both the gram-positive and gram-negative bacterial infection so as to protect fish from getting infected [18,27].

PTX domain is a conservative motif. Such as the humans’ SAP, OnSAP also has calcium-binding sites; each monomer could bind 2 Ca^2+^, and the amino acid of these binding sites are the same except for one. In humans, SAP shared 51% sequence homology with CRP, which is also short pentraxin and is the main acute phase protein in humans [28,29,30,31]. Based on this information, we compared the tridimensional structure predicted by Alphafold 2.0 between them, and the result demonstrates that they were highly similar (Figure 3H). Moreover, the PTX domain of long-pentraxin OnPTX3 is also the same as OnSAP. Structure analysis told us these above-mentioned pentraxins might share similar functions, and therefore, due to the extra domain sequence named N-terminal region of PTX3, different functions would be found between OnPTX3 and OnSAP (data unpublished). Pentraxin is a kind of protein that is formed with five of the same monomers via non-covalent bonds. Moreover, recombinant protein demonstrated abilities in bacteria combination and agglutination in the presence of Ca^2+^, which strongly demonstrated that PTX domain is the critical part of pathogen identification [6].

The function of SAP regulating phagocytosing is a typical phenomenon among many species, from teleost to mammals. For example, SAP-opsonizing zymosans could be phagocytosed by human macrophages through FcγRIIa [5]. Moreover, SAP plays an important role in assisting phagocytosis towards the *Streptococcus pneumoniae* through the classical pathway of complement system instead of interacting with FcγR directly [9,23]. In teleost fish, SAP-treated *Pseudomonas fluorescens* enhanced the phagocytosis of half-smooth tongue sole [2]. In this study, we demonstrated that OnSAP was able to promote the MO/Mø phagocytosis of *S. agalactiae* and *A. hydrophila.* Based on the above result, OnSAP acts as a PRRs in early vertebrate: OnSAP could bind bacteria and then interact with the cell surface phagocytosis receptor, perhaps Fc Receptor, which might be important for early vertebrate resistance to foreign invasion.

In this study, we demonstrated the function of the OnSAP regulating complement system. CRBCs were employed for the hemolysis assay (Figure 4H). (r)OnSAP could significantly improve the lysis rate. SAP, which has the ability to interact with C1q, MBL and C4BP, is a regulator that assists different complement molecules [11,14]. On the one hand, the classical pathway could be activated through SAP binding antigen, resulting in recruiting C1 complexes to the target surface [9,28]. On the other hand, in humans, MBL combines with SAP with its CLR domain and formed a complex that could amplify complement activation; this process may circumvent MSAP (MBL-associated serine protease) [6]. However, such studies are still rare in bony fish that are necessary to explore the relationship among OnSAP, OnMBL and OnMASP or OnSAP and OnC1q.

In conclusion, the functional characterization of SAP in Nile tilapia was explored and identified in this study. First, the transcript of OnSAP was found to express widely in many tissues and its expression was up-regulated significantly after in vivo stimulation with bacterial pathogen. Further, recombinant OnSAP possessed the capabilities to bind and agglutinate bacteria, and even promoted phagocytosis bacteria cells by monocytes and macrophages. Moreover, the OnSAP could enhance complement-mediated hemolysis. To sum up, our result enriched the function of SAP in a primary vertebrate, which could help us to better understand the biological functions of the pentraxin family.

## 4. Materials and Methods

### 4.1. Animal Collection, Bacterial Challenge and Sample Preparation

Nile tilapia weighing 80 ± 10 g were harvested from a Tilapia Cultural Farm in Guangzhou, Guangdong, China. Upon arrival, fish were cultured in the automatic filtering aquaculture system with a stocking rate of 10 g/L under 28 ± 2 °C for three weeks [32]. All animal experiments were conducted under a protocol approved by the University Animal Care and Use Committee of the South China Normal University (an approval reference number SCNU-SLS-2021-012) [32].

To explore the expression of SAP in healthy fish, eleven tissues such as the liver, head kidney, hind kidney, muscle, heart, spleen, skin, gills, intestine, peripheral blood and thymus were harvested after being anesthetized with 0.02% MS-222 (tricaine methane sulfonate, Aladin, Shanghai, China). Liquid nitrogen was immediately applied to freeze them and later stored at −80 °C for further use [33]. Fresh serum of peripheral blood was collected after centrifugation at 500× *g* for 10 min and was stored at −20 °C for further use [34].

The fish were immunized with 100 μL (1 × 10^6^ CFU/fish, 1 × 10^7^ CFU/mL concentration) live bacteria liquid such as *S. agalactiae* (ZQ1901) or *A. hydrophila* (BYK00810) dissolved in PBS (10 mM phosphate, 150 mM NaCl, pH 7.4), and grouped in stimulation groups, respectively [35]. As for the control group, PBS of equal volume was immunized [25,36,37,38].

### 4.2. RNA Extraction and cDNA Synthesis

Total RNA from the healthy fish liver was extracted using Trizol Reagent (Vazyme, Nanjing, China) for amplifying the ORF of tilapia SAP. After a successful acquisition of total RNA, the template was synthesized with the HiScript^®^ II 1st Strand cDNA Synthesis Kit (Vazyme, Nanjing, China). Similarly, the templates of every sample collected before were synthesized using the same protocol and then they were stored at −80 °C for further analysis [36].

### 4.3. Gene Cloning and Sequence Analysis

Complete ORF of *OnSAP* gene was cloned according to the predicted sequence of *Oreochromis niloticus* SAP mRNA (GenBank accession LOC109196457) based on the primer devised by software Primer Premier 5.0 (Appendix A). After being tested by a 1% agarose gel electrophoresis (Bio-Rad, Hercules, CA, USA), the PCR products were ligated into the pMD-18T vector (TaKaRa, Kyoto, Japan), transforming into competent *E. coli* cells (DH5α, TaKaRa, Kyoto, Japan).

Multiple amino acid sequence alignments of OnSAP analyzed by ExPASy tools (http://expasy.org/tools (accessed on 25 May 2021)) were achieved by the DNAMAN software and Clustalw2 program (http://www.ebi.ac.uk/Tools/clstalw2 (accessed on 25 May 2021)). Using the neighbor-joining method with 1000 bootstrap replications, the MEGAX was employed to give a phylogenic tree [33].

### 4.4. Quantitative Real-Time PCR (qRT-PCR)

The constitutive OnSAP expressions including healthy and bacterial injection situations were determined by qRT-PCR whose total volume was 20 μL containing 10 μL 2 × TaKaRa Ex Taq™SYBR premix, 3 μL of diluted cDNA (100 ng/μL), 2 μL of each primer shown in Appendix A (2 μM), 2.6 μL DEPC treated water (Vazyme, Nanjing, China), and 0.4 μL Rox Reference Dye II (TaKaRa, Kyoto, Japan). *β*-actin was employed as an internal control to normalize the relative OnSAP expression level, whose fold changes was calculated using the 2^−ΔΔCt^ method [33,34,36,39]. The qRT-PCR program follows the description of the published works [40].

### 4.5. Isolation and Culture of Monocytes/Macrophages In Vitro

Head kidneys were collected from healthy *O. niloticus* weight 200 ± 10 g and grinded into cellular homogenate using a 1 mL disposable sterilized syringe in a cell culture dish with 10 mL RPMI 1640 medium. Different concentrations of percoll could be selected to further isolate MO/Mø cells as described. Diluted with 100 mM PBS in a ratio of 9:1, the processed original percoll solution (Sigma-Aldrich, Shanghai, China) could be used to further adjust the concentration to 54% and 31% using 10 mM PBS. Then, the separation liquid was prepared using 10 mL 54% percoll covered with 10 mL of 31% percoll. After slightly adding head kidney MO/Mø cell suspensions of Nile tilapia to separation liquid and centrifuging, the MO/Mø fraction could be collected from the 31–54% interface of percoll [38].

After being diluted with cell culture (10% FBS + 1% penicillin-streptomycin), the cell concentration was appraised by 0.4% trypan blue exclusion in order to control the density to 5 × 10^6^ cells/mL. The above system was incubated and cultured in 96-well plates (100 μL/well) (Thermo Fisher Scientific, Waltham, MA, USA) at 25 °C for 3 days. Removing non-adherent cells, the MO/Mø were washed with 10 mL RPMI 1640 basic medium 3 times and resuspended. Finally, they were cultured in the 96-well plates as mentioned above for further experiment [41].

### 4.6. Construction of Recombinant OnSAP Plasmid, Expressing and Purification

Amplified by the specific primers EOnSAP-F and EOnSAP-R using *EcoR I* and *Hind*
*III* as restriction sites, the PCR products were inserted into the pMD-18T vector. Later, recombinant pMD-18T plasmid and pET-32a digested with *EcoR* I and *Hind* III were the further form expression plasmid pET-32a-SAP, which was transferred into BL21 (DE3) (Tiangen, Beijing, China), and cultured in LB-ampicillin. Induced at 37 °C for 6 h after adding Isopropyl-*β*-D-thiogalactopyranpside (IPTG) to a final concentration of 1 mM, once the culture medium reached an O.D. 600 of 0.6–0.8, the culture medium was centrifuged at 3300× *g* for 30 min at 4 °C. PBS was employed to re-suspend the cells. Next, lysozyme (dissolved in water to 10 mg/mL; Sigma-Aldrich, Shanghai, China) was added to the resuspended cells at a ratio of 1:100 and kept the reaction for 3 h. After sufficient reaction, above cells mixture was disrupted by an Ultrasonic Processor (ShunmaTech, Guangzhou, China) to release the protein. Then, cell lysate was centrifuged at 10,000 rpm for 30 min at 4 °C, and the precipitate was re-suspended by Lysis Buffer (8 M Urea, 50 mM NaH_2_PO_4_, 300 mM NaCl, 10 mM imidazole, pH8.0). Ni-NTA His Band Resin columns (Novagen, Darmstadt, Germany) were applied to purify following the protocol [19]. The columns with the nicker ions were balanced by 5 mL Lysis Buffer and filled with protein solution for full contact. After the liquid in the column is drained out, 15 mL Wash Buffer (8 M Urea, 50 mM NaH_2_PO_4_, 300 mM NaCl, 20 mM imidazole, pH 8.0) and 5 mL Elution Buffer (8 M Urea, 50 mM NaH_2_PO_4_, 300 mM NaCl, pH8.0) with different concentrations of imidazole (20 mM, 40 mM, 60 mM, 200 mM and 1 M) were added in sequence and collected respectively to obtain the recombinant protein. Obtained protein was concentrated by PEG 20,000 and separated by 12% SDS-PAGE gel Electrophoresis.

### 4.7. Tridimensional Structure Analysis of OnSAP

Tridimensional structure of OnSAP was predicted by Alphafold 2.0, which was deployed on the Ubuntu 16.04.7 lts (GNU/linux5.4.0-67-generic x86_64) GPU server equip with Tesla v100 32 GB [20]. The human SAP photographed and painted by X-RAY DIFFRACTION 2.2 Å was obtained from RCSB protein data bank (PDB) (https://www.rcsb.org/ (accessed on 16 December 2021), PDB identity number: 1GYK). Moreover, the structures of OnCRP (Gene symbol LOC109204179) and OnPTX3 (XP_005474584.1) were also predicted by AlphaFold 2.0 for comparison. Further the similarity among OnSAP, Human SAP and CRP was compared by Pymol 2.5.

### 4.8. Western Blotting

Being resolved in 12% SDS-PAGE gel electrophoresis, recombinant OnSAP proteins (with or without *β*-2-ME processing) were transferred onto nitrocellulose membranes. Washed three times with 1 × TTBS (50 mmol/L Tris-HCl, 150 mmol/L NaCl, pH = 7.4) containing 0.05% (*v*/*v*) Tween-20 (PBST), mouse against 6 × His tag IgG monoclonal antibodies (Bio Basic, Markham, On, Canada) (1:1500 dilution) and mouse polyclonal antibodies (1:500 dilution) were employed as the primary antibody for 1 h incubation at 37 °C. As for mouse polyclonal antibodies against recombinant OnSAP, briefly, six-week-old female BALB/c mice (purchased from Guangdong Animal Experiment Center (Guangzhou, China)) were intraperitoneally immunized with (r)OnSAP proteins 50 μg (100 μL) mixed in complete Freund’s adjuvant (Sigma-Aldrich, New York, NY, USA). The immunized mice were bled from the tail vein on the fifth day and boosted every ten days using incomplete Freund’s adjuvant (Sigma-Aldrich, Darmstadt, Germany) mixed OnSAP proteins 25 μg (100 μL). The serum containing polyclonal antibodies was collected from mice blood and stored at −80 °C [42]. After washing, the HRP-conjugated goat anti-mouse IgG (1:20,000 dilution) (Bio Basic, Markham, ON, Canada) was used as secondary antibody and incubated for 1 h at 37 °C. Bands in membrane were visualized using a blot scanner (Bio-Rad, Hercules, CA, USA) and analyzed by Image Lab 3.0 (Bio-Rad, Hercules, CA, USA).

### 4.9. ELISA for Analysis of Bacteria Combination with (r)OnSAP

Diluted with coating buffer (15 mM Na_2_CO_3_, 35 mM NaHCO_3_, pH 9.6), a volume of 100 μL/well (1 × 10^7^ CFU/mL) *S. agalactiae* or *A. hydrophila* was coated in the 96-well plates (Costar, 3590, USA) at 37 °C for 1 h. In every step, the plates were washed three times using TBS/Ca^2+^ (2 mmol/L) and the temperature of ELISA was controlled at 37 °C. The (r)Trx and recombinant OnSAP was binded to biotin and diluted to different concentrations (50 μg/mL, 25 μg/mL, 12.5 μg/mL, 6.25 μg/mL, 3.125 μg/mL, 1.5625 μg/mL and 0.5 μg/mL). Each well of plates was blocked at 200 μL. As a control group, the well was filled with 200 μL 1 × TBS/Ca^2+^. After being washed, biotinylated antibodies were employed in combination with streptavidin-HRP (SouthernBiotech, Being, China). ABTS (0.4 mg/mL) (Aladin, Shanghai, China) in substrate buffer with 0.03% H_2_O_2_ was used to develop plates and the optical density rate (O.D.) was measured using a Microplate Reader (Thermo Fisher Scientific, Waltham, MA, USA) at 405 nm [32,43].

### 4.10. Agglutination Assay

After being re-suspended in TBS/Ca^2+^ buffer, bacteria including *S. agalactiae* or *A. hydrophila* (1 × 10^8^ CFU/mL) were washed for four times and heated at 65 °C for 15 min. Afterwards, being labeled with 200 μL (1 mg/mL) fluorescein isothiocyanate (FITC, Sigma, America), the bacteria were lucifugal incubated for 30 min at 25 °C. After being washed four times and re-suspended in TBS/Ca^2+^ buffer, labeled bacteria were mixed with (r)OnSAP (40 μg/mL or 80 μg/mL), (r)Trx (40 μg/mL) as well as TBS buffer, respectively. After incubating for 1 h at 25 °C, treated samples were observed and photographed through a fluorescence microscope (Leica, Heerbrugg, Switzerland) [44].

### 4.11. Assay for Effect of (r)OnSAP on Phagocytosis

To explore the influence of (r)OnSAP on phagocytosis, flow cytometric analysis was employed [11,45]. Briefly, 100 μL FITC-labeled *S. agalactiae* (2 × 10^7^ CFU/mL) or *A. hydrophila* (2 × 10^7^ CFU/mL) was mixed with 100 μL (r)OnSAP (50 μg/mL) in the presence of Ca^2+^ the in dark at 25 °C for 1 h. Then, the above mixture was incubated with 1 × 10^5^ cells of the MO/Mø suspension. Incubated at 25 °C for 1 h, mixture tubes were shaken every 10 min. As for the control, bacteria were similarly treated by TBS or (r)Trx instead of (r)OnSAP. Mixtures after incubation were centrifuged under 100× *g* for 10 min to remove the non-ingested bacteria from the macrophages, washed using 1 × PBS and re-suspended in TBS (pH 7.4). After adding 1 μL of trypan blue (0.4%) 10,000 individual cells of each sample were analyzed in flow cytometer FACS Aria III (BD Biosciences, Franklin Lakes, NJ, USA). Defined as the percentage of the macrophages with one or more engulfed bacteria within the total cell population, the phagocytosis rate was analyzed by Flowjo V10 software [34,36].

### 4.12. Effect of OnSAP on Tilapia Complement System-Mediated Cell Lysis

#### 4.12.1. The (Half Complement System-Mediated Cell Hemolysis Rate) CH50 Assay

Fresh tilapia serum and 2% chicken red blood cells (CRBCs) (SenBeiJia, Nanjing, China) were employed for the CH50 assay. DVB (0.5 mM MgCl_2_, 142 mM NaCl, 5 mM sodium barbital, 0.15 mM CaCl_2_ and 0.5 mM MgCl_2,_ pH 7.5), the buffer for the assay, was applied for serum dilution. First, for the activation of the complement system, 2 μg mannose was added to 1 mL serum (1:10 dilution) and the above-mentioned mixture was incubated at 25 °C for 1 h. Second, 0, 10, 20, 30, 40, 50, 60, 70, 80, 90 and 100 μL serum (1:10 dilution) were added to the tube and then DVB buffer was added to achieve a total of 150 μL of volume. Afterward, the tubes after adding 100 μL CRBCs were incubated at 25 °C for 1 h. Finally, cells were centrifuged at 500× *g* for 5 min at 4 °C. The resultant supernatant of 100 μL was moved to a 96-well microporous plate. Absorbance at O.D. 405 nm was detected by a microplate reader (Thermo Fisher Scientific, Waltham, MA, USA) [46].

#### 4.12.2. Hemolytic Assay

The (r)OnSAP and (r)Trx were diluted to different concentrations (5 μg, 10 μg, and 20 μg) using 1 × TBS/Ca^2+^. Then, the protein and control group TBS were added to activated tilapia serum for 1 h at 25 °C. After incubation, 100 μL CRBCs and DVB were added to make a total volume of 250 μL. The above mixture was also incubated for 1 h at 25 °C and then centrifuged using 500× *g* for 5 min at 4 °C. Similarly, 100 μL of the resultant supernatant was employed for testing O.D. 405 nm absorbance. The calculation of lysis percentage was analyzed as described [46]. All the experiments were performed three independent times.

## Figures and Tables

**Figure 1 ijms-23-09468-f001:**
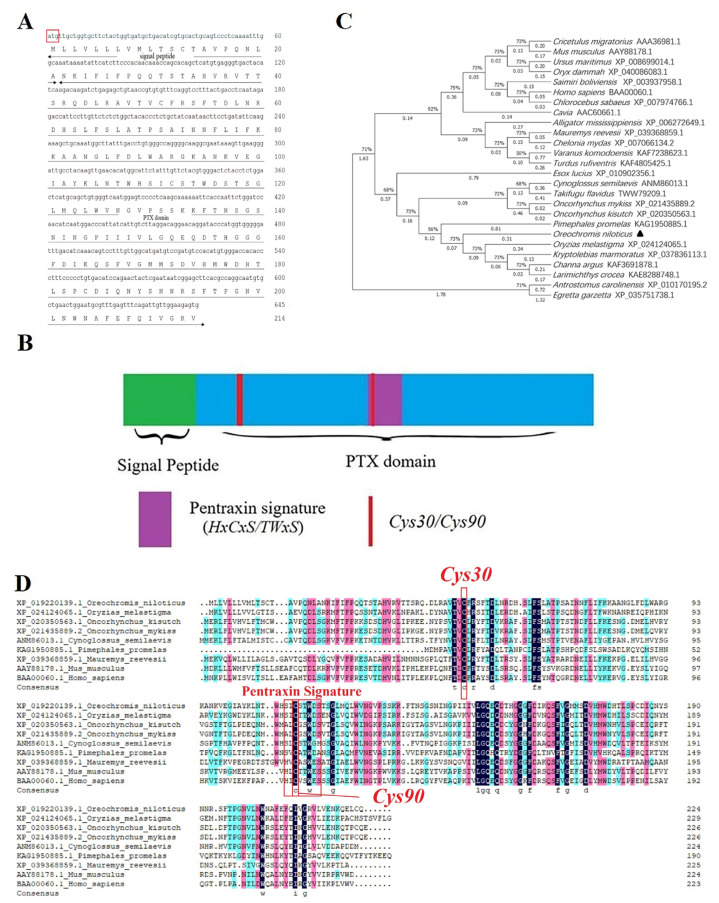
Sequence analysis of OnSAP. (**A**,**B**). Protein sequence analysis of OnSAP. PTX domain occupies the full length of the protein, and a conservative structure named Pentraxin signature is located in the middle of the protein. (**C**). Phylogenetic tree of SAP family members. Numbers at each branch indicated the percentage bootstrap values on 1000 replicates. (**D**). Multiple sequence alignment of the deduced amino acid sequences of SAP among different species. GenBank accession numbers of the used species for comparison are *Homo sapiens* (BAA00060.1), *Mus musculus* (AAY88178.1), *Oryzias melastigma* (XP_024124065.1), *Oncorhynchus kisutch* (XP_020350563.1), *Cynoglossus semilaevis* (ANM86013.1), *Chelonia mydas* (XP_007066134.2), respectively. The conservative motifs were highlighted in the red box, including important domains such as Cys36/Cys90 and the Pentraxin signature.

**Figure 2 ijms-23-09468-f002:**
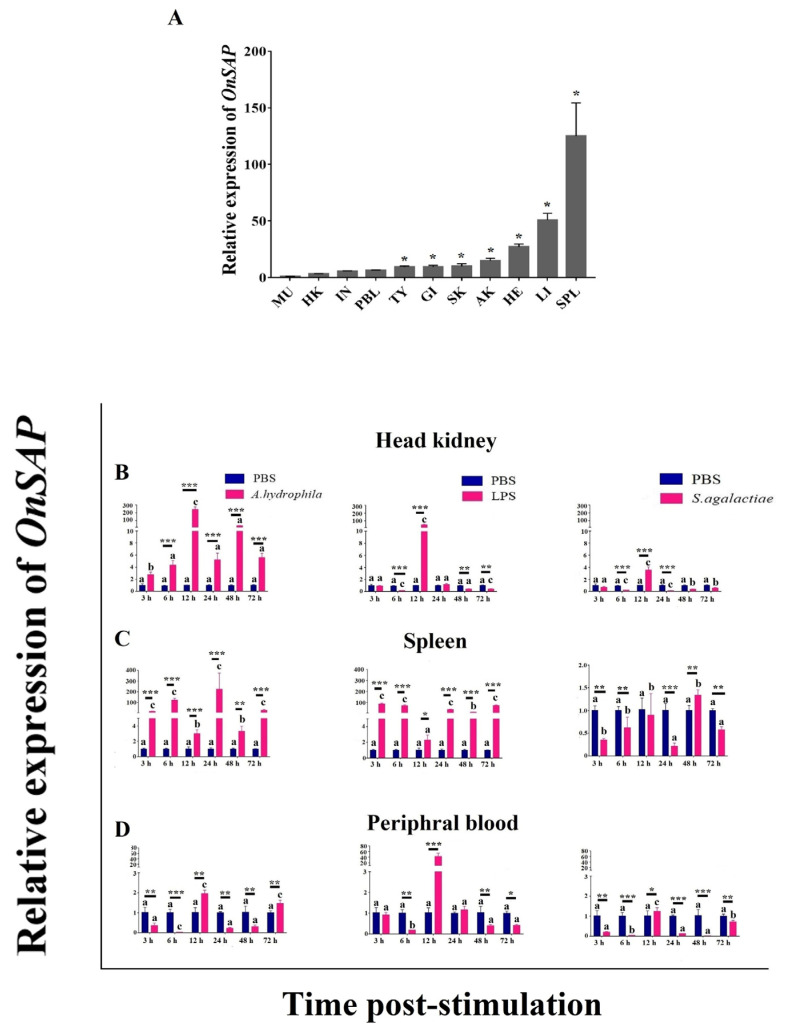
(**A**) Tissue distribution of *OnSAP* mRNA in healthy Nile tilapia. Muscle was regarded as a standard that could make relative comparison with different tissues and the ratio reflect the comparison normalized against *β*-actin. The results were mean ± SD of three replicate samples. (SPL: spleen; LI: liver; HE: heart; AK: head kidney; SK: skin; GI: gill; TY: thymus; PBL: peripheral blood; IN: intestinal; HK: trunk kidney; MU: muscle). Asterisks indicate significant differences (*p* < 0.05) compared to muscle. Temporal mRNA expression of *OnSAP* transcript after *S. agalactiae* (1 × 10^7^ CFU/mL), *A. hydrophila* (1 × 10^7^ CFU/mL) and LPS stimulation in the head kidney (**B**), spleen (**C**) and peripheral blood (**D**). Normalization of mRNA level of OnSAP was referred to *β*-actin and fold units that were calculated deciding the values of the PBS vaccinated tissues. Different letters (a, b, c and *, **, ***, *p* < 0.05) were calculated by *t*-test, reflecting the statistical significance among different time points compared with 3th h (a, b, c) or between groups (*, **, ***).

**Figure 3 ijms-23-09468-f003:**
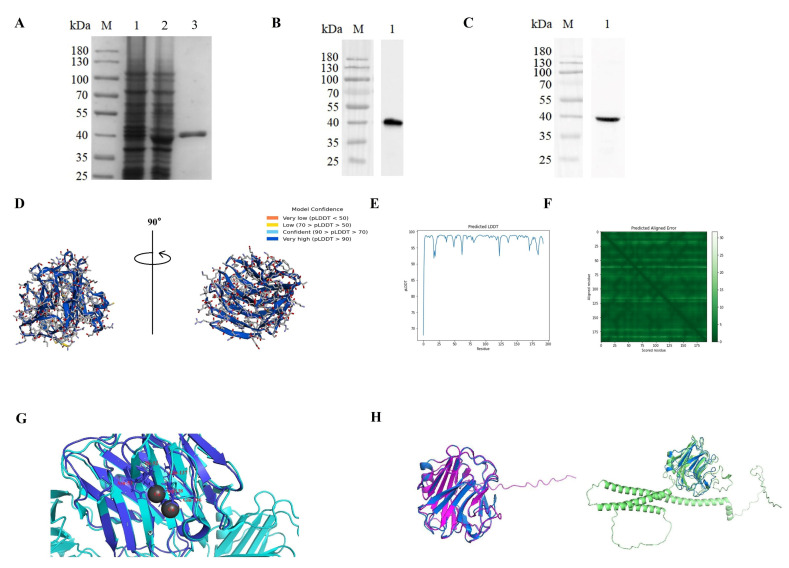
Purification of (r)OnSAP and Western blot analysis. (**A**) Lane M, markers; Lane 1, the bacteria liquid without IPTG induction; Lane 2, bacteria liquid was induced with 1 mM IPTG at 37 °C for 6 h; Lane 3, purified (r)OnSAP fusion protein. (**B**,**C**) Western blot analysis of (r)OnSAP using anti-His tag mouse monoclonal antibody or polyclonal mouse anti-(r)OnSAP antibody as the primary antibody. (**D**) Tridimensional structure of OnSAP predicted by AlphaFold 2.0. (**E**) pLDDT of OnSAP remained above 90 at the start of the prediction. (**F**) PAE stayed below 10. (**G**) The similarity between OnSAP (blue) and humans SAP (sky-blue) monomer. The roots’ mean square displacement (RMSD) is 0.942 computed by Pymol 2.5. The calcium-binding sites were labeled in red words. (**H**) Similarity between OnSAP and OnCRP (pink)/OnPTX3 (green), which are also predicted by AlphaFold 2.0. The RMSD is 0.33 and 1.528.

**Figure 4 ijms-23-09468-f004:**
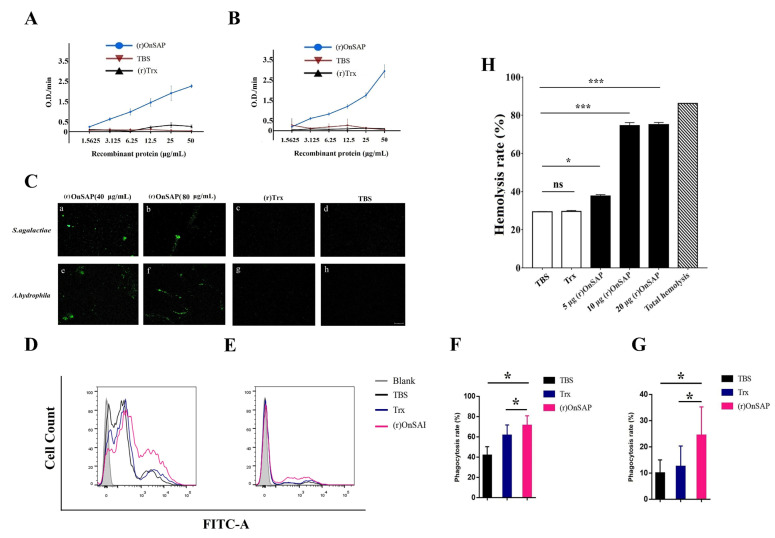
ELISA result of the combination between OnSAP among (**A**) *S. agalactiae* and (**B**) *A. hydrophila*. (**C**) Observation of binding and agglutinating of (r)OnSAP to *S. agalactiae* and *A. hydrophila* using a fluorescence microscope (Objective 10-fold). FITC-labeled *S. agalactiae* and *A. hydrophila* were incubated with different concentrations of (r)OnSAP (a, b, e, f), pET-32a (c, g) and TBS control (d, h), respectively. Effects of (r)OnSAP on phagocytosis. (**D**,**E**) The histogram of flow cytometric analyses of MO/Mø phagocytosing *A. hydrophila* and *S. agalactiae* pre-treating with TBS, (r)Trx and (r) OnSAP. (**F**) Phagocytic percentage of MO/Mø phagocytosing *A. hydrophila.* Pre-incubated with different groups. (**G**) Phagocytic percentage of MO/Mø phagocytosing *S. agalactiae* pre-incubated with different groups. (**H**) Enhancement of tilapia complement-mediated cell lysis by (r)OnSAP. Different pre-treated normal tilapia serum was incubated with CRBCs at 25 °C for 1 h. The results are representative of three independent experiments. Different letters (ns, *, ***, *p* < 0.05) were calculated by *t*-test, reflecting the statistical significance.

## Data Availability

The data presented in this study are available from the corresponding author on reasonable request.

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
