# Peer review of "Functional Characterization of Serum Amyloid P Component (SAP) in Host Defense against Bacterial Infection in a Primary Vertebrate"

_ijms, 2022, doi:10.3390/ijms23169468_

Round 1
Reviewer 1 Report
Dear Authors,
Please see attached comments.

Author Response
Response to reviewers.
Comments to the Author (in italics) with our responses (blod script)
We believe that we have accommodated the requests of the reviewers and in doing so, the quality of the manuscript has been greatly improved.
Reviewer #1
The authors conducted all experiments needed for molecular biology research: cloning OnSAP cDNA, analyzing molecule sequences, simulating 3D structure with the latest program-AF2, examining mRNA expression in different tissues, recombinantly expressing and purifying OnSAP protein from bacteria, as well as analyzing protein function by detecting the binding activity of rOnSAP to different species of bacteria.
Overall, the manuscript is well structured, and the work has been carried out to a competent standard concerning molecular biology. However, there are some major issues. Also, the authors should pay attention to their sloppy grammar and typo in places:
- Lines 86-90: the description of this part is not aligned with Fig. 1D. Please review, revise, and improve the resolution of Fig. 1, particularly Fig. 1A.
We appreciate the reviewer’s comment and completely agree with the suggestion. Following the suggestion, the description has been corrected in line 88-89 in the revised manuscript. Besides, the resolution of Figure 1 has been improved in the revised manuscript.
Line 88-89 (Modified in the revised manuscript): OnSAP shares 56.73%, 47.98%, 47.27%, 36%, 38.12%, 37%, and 37.37% sequence similarity with Oryzias melastigma SAP, Oncorhynchus kisutch SAP, Cynoglossus semilaevis SAP, Chelonia mydas SAP, Mauremys reevesii SAP, human SAP and mouse SAP respectively, revealing its evolutionary conservation.
- Lines 92-95: How do you define “clustered together”? For example, are M. musculus and H. sapiens, or O. niloticus and O. kisutch in the same group in Fig. 1C?
We appreciate the reviewer’s comment. We agree with that the way using the root of the phylogenetic tree as a branch is not rigorous. Following the valuable suggestion, a more suitable way has been applied to explain the phylogenetic tree in a bottom-up manner in line 93-99 in the revised manuscript.
Line 93-99 (Modified in the revised manuscript): SAP of O. niloticus firstly was clustered with Oryzias melastigma, Kryptolebias marmoratus et al., and then clustered with another branch including O. kisutch, O.mykiss et al., which formed a branch representing bony fish. Similarly, SAP of H. sapiens was clustered with Chlorocebus sabaeus, a primate, and this branch with another branch including M. musculus, Cricetulus migratorius formed a new branch representing mammals. Above results reflect the evolutionary route of the SAP protein.
- Lines 105-106: Any explanation why it is downregulated after treating with S. agalactiae.
We understand the reviewer’s concern that the OnSAP expression was downregulated after treating with S. agalactiae.
Bacterium Streptococcus iniae has a close relationship with S. agalactiae. In the study of rock bream, expression of rock bream SAP (RbSAP) was observed to be down-regulated after Streptococcus iniae infection [1]. Further, the RbSAP protein could bind and agglutinate the gram-positive bacterium S. iniae. The above findings in the study of the RbSAP are in line with the data in our study. However, the mechanism to cause the phenomenon remains unclear, which needs further investigation.
[1] Choi K, Sang H, et al. Functional characterization and expression analysis of recombinant serum amyloid P isoform 1 (RbSAP1) from rock bream (Oplegnathus fasciatus). Fish Shellfish. Immun., 2015, 45, 277-285.
- Figure 4A &B: Not sure if expressed the mature OnSAP, but the sizes of recombinantly produced protein in stained gel and in Western blotting are not the same size. Please explain, why?
We appreciate the comment and understand the reviewer’s concern that the sizes of recombinant protein in stained gel and in Western blotting may not be the same size.
We believe that the difficulty was caused due to the different buffer of the recombinant protein. In the Western blotting analysis, the eluate protein was employed without removement of urea in order to check protein purity. Urea is widely used in the purification of recombinant proteins. Urea plus SDS is able to achieve a more complete denaturation of polypeptides, which leads to faster electrophoretic mobility [1].
To solve this difficulty, the urea-removed protein has been employed in the Western blotting analysis. The Western blotting results are shown as follows: the left figure is Western blotting analysis using mouse against 6 × His tag IgG monoclonal antibodies (BBI, USA) (1:1500 dilution), and the right figure is analyzed using mouse anti-tilapia SAP polyclonal antibodies (1:500 dilution). The data show that the protein size matches the size prediction, which has been provided in the Figure 3 in the revised manuscript.
[1] Abraham G, Cooper PD. Anomalous behavior of certain poliovirus polypeptides during SDS-gel electrophoresis. Anal Biochem. 1976, 73(2), 439-46.
- Minors, for example:
1) Lines 14-16: “In this study, the expression and functional characterization of SAP (OnSAP) in Nile tilapia (Oreochromis niloticus), a primary vertebrate, were investigated”
Line 14-16 (Modified in the revised manuscript): In this study, the expression and functional characterization of SAP (OnSAP) in Nile tilapia (Oreochromis niloticus), a primary vertebrate, were investigated.
2) Line 18: “…is highly similar to those of humans”
Line 18 (Modified in the revised manuscript): those of humans
3) Line 83: “…and encodes for 214…”
Line 84 (Modified in the revised manuscript): encodes for
4) Lines 86- 87: “…among sequences…”
Line 88 (Modified in the revised manuscript): among sequences
5) Line 121: Root Mean Squared (RMS) Displacement.
Line 133 (Modified in the revised manuscript): Root Mean Squared (RMS) Displacement
6) Lines 135-136: “…could be photographed…”?
Line 147 (Modified in the revised manuscript): was detected
7) Line 173: “… was highly expressed…”?
Line 185 (Modified in the revised manuscript): was highly expressed
8) Line 185: “…was significantly increasing…”?
Line 197 (Modified in the revised manuscript): significantly increased
9) Line 197-198: “…these 3 kinds of pentraxin…”?
Line 210 (Modified in the revised manuscript): above-mentioned pentraxins
10) Line 256: “-80 °C”
Line 270 (Modified in the revised manuscript): -80°C
11) Line 270: please show how much cDNA (μg or ng) in 3 μL of diluted cDNA.
Line 285 (Supplemented in the revised manuscript): The information of 100 ng/μl diluted cDNA has been provided in the revised manuscript.
12) Line 303: Tridimensional structural analysis of OnSAP.
The “3-D” has been modified to “Tridimensional” in the revised manuscript.
13) Line 307: Please show the PDB identity number.
Line 335 (Supplemented in the revised manuscript): The PDB identity number of 1GYK has been provided in the revised manuscript.
14) Line 310: “compared’, instead of “calculated”?
Line 338 (Modified in the revised manuscript): compared

Reviewer 2 Report
In the present manuscript, Li et. el. have studied, expressed and functionally characterized OnSAP. Serum amyloid P component (SAP) is a short pentraxin which plays a key role in innate immune defence. The expression of OnSAP was detected and found in multiple tissues of NILE tilapa. The functional assays with r-OnSAP revealed the bacterial binding, agglutinating activities of OnSAP. It was also found to promote phagocytosis of bacteria by macrophages and monocytes
Authors did a appreciable job in studying the SAP in fishes but unfortunately, there is large scope of improvement in the present form of manuscript. Following points could be considered.
In Figure 2, the relative expression of OnSAP in tissues should be analyzed statistically with p values shown in figure.
In Figure 3 the basal expression of OnSAP is relatively higher in spleens in part B, could authors explain these high values in spleen tissues, as these values are only shown in S. agalactiae graph. The spleen tissue when treated with PBS should result in same expression level as shown in comparison with LPS and A. hydrophilla . Along with the pair wise comparisons, authors should also analyze the obtained results by performing multiple group comparisons to evaluated the significance among different time points.
Figures 2 and 3 could be combined
In results while describing about recombinant expression of OnSAP authors could also mention about the chromatography used for purification of recombinant protein, authors should modify the description, as Trx-pET-32A is an expression vector and it is used for the expression of recombinant OnSAP.
The expressed protein is confirmed by use of anti-6 His-tag mouse 113 monoclonal antibody, which does not ensure the expression of complete protein, can authors repeat the confirmation using antibodies against OnSAP.
There is some rearrangement of figures and text is required, Figure 5 could be merged with Figure 4A and Figure 4B. With the current arrangement readers have to go back and forth with text and figures.
While describing the tertiary structure of OnSAP, authors could add more description about pLDDT, the relevance of scores with accuracy of prediction could be added. Authors could also add the verification of 3D structure prediction in supplementary data such as Ramachandran PLOT analysis.
Figure 4E the scale is missing on microscopic figures,
In Figure 6, have authors compared the Trx group and OnSAP group for the significant difference, the OnSAP group is needed to be statisticaly compared with TRX group in C and D (Figure 6).
Figures 4D, 4E, 6 and 7 can be combined together in a single figure. As all these figures are exhibiting the functional aspects if OnSAP.
The raw CT values of OnSAP expression in tissues are required to be provided in supplementary material.
Table1 could be moved to supplementary material.
As the difference of functions between OnPTX3 and OnSAP are still unclear, while discussing the results the use of “obvious” word should be refrained.
In methods, the section describing the purification of protein in detail is required to be added.
Author Response
Response to reviewers.
Comments to the Author (in italics) with our responses (red script)
We believe that we have accommodated the requests of the reviewers and in doing so, the quality of the manuscript has been greatly improved.
Reviewer #2
In the present manuscript, Li et al. have studied, expressed and functionally characterized OnSAP. Serum amyloid P component (SAP) is a short pentraxin which plays a key role in innate immune defence. The expression of OnSAP was detected and found in multiple tissues of Nile tilapia. The functional assays with r-OnSAP revealed the bacterial binding, agglutinating activities of OnSAP. It was also found to promote phagocytosis of bacteria by macrophages and monocytes. Authors did an appreciable job in studying the SAP in fishes but unfortunately, there is large scope of improvement in the present form of manuscript. Following points could be considered.
Major issues
1. In Figure 2, the relative expression of OnSAP in tissues should be analyzed statistically with p values shown in figure.
We appreciate the reviewer’s comment and completely agree with the valuable suggestion. Following the suggestion, the statistical analysis in the Figure 2, based on the methods of the published paper [1], has been performed and provided in the revised manuscript.
[1] Choi K, Sang H, et al. Functional characterization and expression analysis of recombinant serum amyloid P isoform 1 (RbSAP1) from rock bream (Oplegnathus fasciatus). Fish Shellfish. Immun., 2015, 45, 277-285.
In Figure 3 the basal expression of OnSAP is relatively higher in spleens in part B, could authors explain these high values in spleen tissues, as these values are only shown in S. agalactiae graph. The spleen tissue when treated with PBS should result in same expression level as shown in comparison with LPS and A. hydrophilla. Along with the pair wise comparisons, authors should also analyze the obtained results by performing multiple group comparisons to evaluate the significance among different time points.
We appreciate the reviewer’s comment and valuable suggestion.
Actually, the basal expression of OnSAP in spleens is around 1 in Figure 2 part B, which expression is similar to that in spleens after S. agalactiae infection. To avoid the misunderstanding, all of the figures in Figure 2, except part C S. agalactiae graph, have been photographed in the same scale.
Following the valuable suggestion, the significances among different time points have been evaluated and marked in the figures with “a, b, c”. Expression of each time point is compared with the earlier time point, and the expression of OnSAP shows the similar significance at the same time point after LPS and A. hydrophilla challenge. Since LPS is component of the cell wall of gram-negative bacteria A. hydrophilla, the results are theoretically reasonable.
- In results while describing about recombinant expression of OnSAP authors could also mention about the chromatography used for purification of recombinant protein, authors should modify the description, as Trx-pET-32A is an expression vector and it is used for the expression of recombinant. The expressed protein is confirmed by use of anti-6 His-tag mouse 113 monoclonal antibody, which does not ensure the expression of complete protein, can authors repeat the confirmation using antibodies against OnSAP.
We appreciate the reviewer’s helpful suggestion. The description about the chromatography used for the purification of recombinant protein has been modified in line 113-114 in the revised manuscript. Besides, following the suggestion, the mouse anti-OnSAP polyclonal antibodies have been employed in the study to ensure the expression of the recombinant OnSAP as the primary antibody, and the information has been provided in line 117 and 339-346 and shown in in Figure 3 part C in the revised manuscript.
Line 113-114 (Modified in the revised manuscript): Recombinant OnSAP protein ((r)OnSAP) was purified following the protocol of Ni-NTA Agarose Resin (Novagen, Germany).
Line 117 (Supplemented in the revised manuscript): …or polyclonal mouse antibody…
Line 344-355 (Supplemented in the revised manuscript): As for mouse polyclonal antibodies against recombinant OnSAP, briefly, six-week-old female BALB/c mice (purchased from Guangdong Animal Experiment Center (Guangzhou, China)) were intraperitoneally immunized with (r)OnSAP proteins 50 μg (100 μL) mixed in complete Freund's adjuvant (Sigma-Aldrich, Germany). The immunized mice were bled from the tail vein on the fifth day and boosted every ten days using incomplete Freund's adjuvant (Sigma-Aldrich, Germany) mixed OnSAP proteins 25 μg (100 μL). The serum containing polyclonal antibodies was collected from mice blood and stored at -80℃ [45]. After washing, the HRP-conjugated goat anti-mouse IgG (1:20000 dilution) (BBI CO., LTD, China) were used as secondary antibody and incubated for 1 h at 37℃. Bands in membrane were visualized using a blot scanner (Bio-Rad ChemiDoc MP, USA) and analyzed by Image Lab 3.0 (Bio-Rad, USA).
[45] Li B, Li Y, et al. Identification and functional characterization of CD154 in T cell-dependent immune response in Nile tilapia (Oreochromis niloticus). Fish Shellfish Immun., 2021, 111, 102-110.
- While describing the tertiary structure of OnSAP, authors could add more description about pLDDT, the relevance of scores with accuracy of prediction could be added. Authors could also add the verification of 3D structure prediction in supplementary data such as Ramachandran PLOT analysis.
We appreciate and agree with the reviewer’s suggestion. In this study, we have plotted based on the alphafold output file, and the results show that the prediction’s pLDDT remains above 90 at the start of the prediction (Fig. left), which confidence is defined as ‘very high’ when over 90. Whereas, the PAE (Predicted Aligned Error) stays below 10 (Fig. right).
The description about the pLDDT and the confidence of prediction has been provided in line 124-129 in the revised manuscript.
Besides, the Ramachandran PLOT analysis has been finished using UCLA-DOE LAB (saves.mbi.ucla.edu/). The proportion of most favored regions residues is 93%. The descriptions have been provided in line 129-131 and line 550-552 in the revised manuscript.
Line 124-129 (Supplemented in the revised manuscript): (LDDT), which displayed the confidence of per-residue in the range 0-100, is best used for intra-domain confidence. Besides, Predicted Aligned Error (PAE) is an index for determining between domains or between chain confidences [20]. According to these indexes, OnSAP structure prediction is accurate. In details, pLDDT remained above 90 (confidence is defined as ‘very high’ when over 90) at the start of the prediction (Fig. 4E), whereas PAE stayed below 10 (Fig. 4F).
Line 129-131 (Supplemented in the revised manuscript): Supplementary Figure 1 showed Ramachandran PLOT analysis provided by UCLA-DOE LAB (saves.mbi.ucla.edu/), which contains the information that proportion of most favored regions residues is 93%.
Line 550-552 (Supplemented in the revised manuscript): The Ramachandran PLOT analysis has been provided in the Supplementary Figure 1 in the revised manuscript.
Supplementary Figure 1
Figure 1 legend: Ramachandran PLOT analysis for OnSAP structure verification
- Figure 4E the scale is missing on microscopic figures.
In the revised manuscript, the scale bar in the Figure 4C has been provided on microscopic figures. Further, all the pictures have been photographed in the same scale.
- In Figure 6, have authors compared the Trx group and OnSAP group for the significant difference, the OnSAP group is needed to be statistically compared with TRX group in C and D (Figure 6).
We appreciate and agree with the reviewer’s comment suggestion. Based on the valuable advice, the statistical comparison between the OnSAP group and the TRX group in the Figure 4F, 4G has been provided in the revised manuscript.
- Figures 2 and 3 could be combined. There is some rearrangement of figures and text is required, Figure 5 could be merged with Figure 4A and Figure 4B. With the current arrangement readers have to go back and forth with text and fig. Figures 4D, 4E, 6 and 7 can be combined together in a single figure. As all these figures are exhibiting the functional aspects of OnSAP.
We appreciate the reviewer’s comments and completely agree with the valuable suggestion. According to the advices, the original Figures 2 and 3 has been combined as the new Figure 2, the original Figure 5 has been merged with Figure 4A and Figure 4B as the new Figure 3, and the original Figures 4D, 4E, 6 and 7 have been combined together as the new Figure 4 in the revised manuscript.
- The raw CT values of OnSAP expression in tissues are required to be provided in supplementary material. Table1 could be moved to supplementary material.
The raw CT values of OnSAP expression in tissues has been provided in the supplementary material, and the Table 1 has also been moved to the supplementary material in the revised manuscript.
- In methods, the section describing the purification of protein in detail is required to be added.
We agree with the reviewer’s suggestion. The detail information of the protein purification has been added in methods in line 115-116 and 315-327 in the revised manuscript.
Line 115-116 (Modified in the revised manuscript): Recombinant OnSAP protein ((r)OnSAP) was purified following the protocol of Ni-NTA Agarose Resin (Novagen, Germany).
Line 315-321 (Supplemented in the revised manuscript): Next, lysozyme (Dissolved in water to 10 mg/mL; Sigma, Germany) was added to the resuspended cells at a ratio of 1:100 and kept the reaction for 3 hours. After sufficient reaction, above cells mixture was disrupted by an Ultrasonic Processor (SHUNMATECH, China) to release the protein. Then, cell lysate was centrifuged at 10,000 rpm for 30 min at 4℃, and the precipitate was re-suspended by Lysis Buffer (8M Urea, 50 mM NaH2PO4, 300 mM NaCl, 10 mM imidazole, pH8.0).
Line 322-327 (Supplemented in the revised manuscript): The columns with the nicker ions were balanced by 5 mL Lysis Buffer and filled with protein solution for full contact. After the liquid in the column is drained out, 15 mL Wash Buffer (8M Urea, 50 mM NaH2PO4, 300 mM NaCl, 20 mM imidazole, pH8.0) and 5 mL Elution Buffer (8M Urea, 50 mM NaH2PO4, 300 mM NaCl, pH8.0) with different concentrations of imidazole (20 mM, 40 mM, 60 mM, 200 mM, and 1M) were added in sequence and collected respectively to obtain recombinant protein.

Round 2
Reviewer 1 Report
Dear Authors,
Thank you for your work in revising your paper.
Reviewer 2 Report
No Comments